# Identification and Characterization of Novel Founder Mutations in *NDRG1*: Refining the Genetic Landscape of Charcot–Marie–Tooth Disease Type 4D in Bulgaria

**DOI:** 10.3390/ijms25169047

**Published:** 2024-08-21

**Authors:** Derek Atkinson, Teodora Chamova, Ayse Candayan, Kristina Kastreva, Ognian Asenov, Ivan Litvinenko, Alejandro Estrada-Cuzcano, Els De Vriendt, Georgi Kukushev, Ivailo Tournev, Albena Jordanova

**Affiliations:** 1Molecular Neurogenomics Group, VIB Center for Molecular Neurology, VIB, 2610 Antwerp, Belgium; 2Department of Biomedical Sciences, University of Antwerp, 2610 Antwerp, Belgium; 3Department of Neurology, Medical University-Sofia, 1431 Sofia, Bulgaria; 4Department of Pediatrics, Medical University-Sofia, 1431 Sofia, Bulgaria; 5Department of Otorhinolaryngology, Military Medical Academy-Sofia, 1606 Sofia, Bulgaria; 6Department of Cognitive Science and Psychology, New Bulgarian University, 1618 Sofia, Bulgaria; 7Department of Medical Chemistry and Biochemistry, Medical University-Sofia, 1431 Sofia, Bulgaria

**Keywords:** *NDRG1*, Charcot–Marie–Tooth neuropathy type 4D, demyelinating neuropathy, founder mutation, non-Roma ethnicity

## Abstract

Charcot–Marie–Tooth neuropathy type 4D (CMT4D) is a rare genetic disorder of the peripheral nervous system caused by biallelic mutations in the N-Myc Downstream Regulated 1 gene (*NDRG1*). Patients present with an early onset demyelinating peripheral neuropathy causing severe distal muscle weakness and sensory loss, leading to loss of ambulation and progressive sensorineural hearing loss. The disorder was initially described in the Roma community due to a common founder mutation, and only a handful of disease-causing variants have been described in this gene so far. Here, we present genetic and clinical findings from a large Bulgarian cohort of demyelinating CMT patients harboring recurrent and novel variants in the *NDRG1* gene. Notably, two splice-site variants are exclusive to Bulgarian Muslims and reside in ancestral haplotypes, suggesting a founder effect. Functional characterization of these novel variants implicates a loss-of-function mechanism due to shorter gene products. Our findings contribute to a deeper understanding of the genetic and clinical heterogeneity of CMT4D and highlight novel founder mutations in the ethnic minority of Bulgarian Muslims.

## 1. Introduction

Charcot–Marie–Tooth neuropathy type 4D (CMT4D; MIM #601455), also known as hereditary motor and sensory neuropathy-Lom (HMSNL) is caused by biallelic pathogenic mutations in the N-Myc Downstream Regulated 1 (*NDRG1*) gene located on chromosome 8q24.22 [1,2]. *NDRG1* is ubiquitously expressed [3]; however, its abundance in Schwann cells is particularly high [2]. *NDRG1* appears to have various roles in Schwann cell differentiation, trafficking and signaling, myelin sheath biosynthesis [2,4,5], cell cycle regulation [6], protection against cellular stress [7], and exocytosis in mast cells [8].

CMT4D was initially described in the Roma families living in the town of Lom in Bulgaria who carried a common founder mutation (p.R148X) [2]. Follow-up studies also reported families outside Bulgaria with Roma origin carrying the same variant [9,10,11,12,13,14], leading to the idea that this rare CMT subtype could be confined to the Roma population. However, several reports documented additional disease-causing variants in single families with different ethnic origin. These include a duplication of exons 6-8 [15], frameshift variants (p.H247Tfs*74 and p.S317Rfs*4) [16,17], nonsense variants (p.Y79X and p.Q185X) [18,19], splice-site variants (c.537+2_537+10del, c.538-1G>A, c.595-2A>G, and c.944-1G>T) [20,21,22,23], and missense variants (p.L146P and p.R234Q) [24] in families with no Roma origin from Bulgaria, Italy, Turkey, India, Saudi Arabia, and China.

CMT4D is clinically characterized by distal muscle wasting and weakness, foot and hand deformities, tendon areflexia, and sensory loss, appearing in the first decade of life. Sensorineural deafness is a hallmark feature of the phenotype and generally develops during the third decade [1,25]. From an electrophysiological point of view, CMT4D is a demyelinating neuropathy involving severe and early axonal loss. It is characterized by markedly reduced conduction velocities and prolonged distal latencies of the motor nerves (including musculocutaneous, axillary, hypoglossal, and facial nerves), absent sensory nerve action potentials (SNAP), and a blink reflex with an unusual three-component response [26,27]. Another typical electrophysiological finding is an abnormality in brain-stem auditory-evoked potentials with prolonged I–V interpeak latencies, suggesting the involvement of the central pathways [27], observed prior to hearing complaints. Nerve histology seems to be distinct, and shows a depletion in myelinated nerve fibers with small size in the few preserved myelinated fibers and poorly developed onion bulbs [14,28,29].

Here, we describe genetic and clinical findings from a Bulgarian cohort of demyelinating CMT patients that carry recurrent or novel variants in the *NDRG1* gene. Notably, two splice variants identified in this cohort (c.538-1G>A and c.327-2A>G) are exclusively present in patients that belong to a specific ethnic minority in Bulgaria called Bulgarian Muslims. Haplotype analysis suggests that the variants originate from an ancestral founder event. Functional characterization of both novel founder variants shows that they result in a shorter gene product, likely causing a loss of protein function. Our findings expand the clinical and genetic landscape of CMT4D and highlight two novel founder mutations in the *NDRG1* gene.

## 2. Results

### 2.1. Identification of Novel Founder Variants in the NDRG1 Gene

We performed a large-scale cohort screening covering all exons, exon-intron boundaries, and 5′UTR of the *NDRG1* locus. Our cohort consisted of 150 index Bulgarian patients with demyelinating CMT. We identified 15 patients from 11 families with recurrent or novel variants in the *NDRG1* gene (Table 1). These included two families carrying the previously reported [21] splice-site variant (families I and II: c.538-1G>A) in homozygosity, one family with a novel homozygous splice-site variant (family III: c.327-2A>G), and one family having both splice-site variants in compound heterozygosity (family IV: c.327-2A>G and c.538-1G>A). Additionally, patients from six families (families VI, VII, VIII, IX, X, and XI) were shown to harbor the recurrent homozygous (p.R148X) founder mutation, and one family (family V) harbored the p.R148X mutation together with a novel frameshift variant (c.508insG, p.E170Gfs*35) in compound heterozygosity. All seven families carrying the founder p.R148X mutation were reported to have Roma origin. Segregation analysis was performed on the available family members, and all of the variants co-segregated with the disease phenotype in the corresponding pedigrees (Appendix A).

The two splice-site variants (c.327-2A>G and c.538-1G>A) were absent from the gnomAD database v3.1.2 and had a SpliceAI prediction of strong splice altering (>0.99). The novel frameshift variant (c.508insG, p.E170Gfs*35) in case V-1 was also absent from the gnomAD database v3.1.2. All of the novel variants identified in this study fulfill the criteria PVS1, PS3, and PM2 and are classified as pathogenic according to the guidelines published by American College of Medical Genetics and Genomics [30].

Interestingly, the splice-site variants c.538-1G>A and c.327-2A>G were identified in a homozygous or compound heterozygous state in four families (families I, II, III, and IV) that belong to another ethnic minority in Bulgaria, the Bulgarian Muslims. We performed a haplotype analysis to assess if these variants derived independently or resulted from a single ancestral mutation in the patients. The c.538-1G>A variant was previously reported by Hunter and colleagues [21], and we were able to obtain biological samples from this family (case I-1). The c.538-1G>A variant resided on a shared haplotype of 2.4 Mb on chromosome 8 between families I, II, and IV, while the c.327-2A>G variant resided on a 9.6 Mb haplotype on chromosome 8 shared between families III and IV (Figure 1). These findings illustrate that both splice-site variants originate from an ancestral founder mutation, similar to the p.R148X mutation described previously in patients with Roma descent.

### 2.2. Functional Impact of Novel NDRG1 Variants

To assess the consequence of the novel splicing events, we initially performed a splice-site prediction with the Alamut Visual Plus software (v1.5.1), integrating different prediction methods, such as SpliceSiteFinder, MaxEntScan, NNSPLICE, and GeneSplicer. This analysis suggested a complete loss of natural splice acceptor sites for both intron 5 (in case of c.327-2A>G) and intron 8 (for c.538-1G>A), with a likely skipping of the adjacent exon. Additionally, the c.538-1G>A variant is predicted to result in the activation of a cryptic splice acceptor site 7bp downstream of the canonical splice acceptor site in exon 9 (c.545), resulting in a frameshift and premature stop codon (Appendix A). Both the skipping of exon 6 (63 nucleotides) and exon 9 (57 nucleotides) was predicted to cause an in-frame deletion and possibly a shorter protein. To confirm the impact of altered splicing, we performed a cDNA analysis on biological material (fresh blood or lymphoblasts) available from cases II-1, II-2, III-1, IV-1, and IV-3. Primers were designed in a way to amplify a cDNA fragment encompassing the neighboring exons to cover the variant and reveal the potential exon skipping (Appendix A).

Analysis of the reverse transcription PCR products on agarose gel resulted in several DNA fragments of different sizes, both for homozygous and compound heterozygous c.327-2A>G and c.538-1G>A variants (Figure 2). The c.327-2G allele resulted in a single shorter PCR fragment compared to the control, while the c.538-1A allele resulted in two shorter fragments. The individuals carrying the compound heterozygous splice-site variants showed an additional PCR fragment at the reference size originating from the second allele. The formation of a heteroduplex band in the reverse transcription PCR products was confirmed by the cleavage of mismatched bases with T7 endonuclease (Appendix A). Agarose gel extraction and sequencing of the PCR fragments revealed that the c.327-2G allele resulted in two transcripts: one allele with skipping of exon 6 and another allele with an activated alternative splice acceptor site 25 bp downstream in exon 6. Additionally, the c.538-1A variant resulted in two transcripts leading to the skipping of exon 9 and the activation of an alternative splice acceptor site 7 bp further, as predicted by the Alamut software (v1.5.1). To assess the splicing preference, transcript ratios were determined using PCR amplification using labeled primers. Quantification of the different alleles was performed over three independent experiments. For both splice-site variants, exon skipping was favored over the adoption of alternative splice acceptor sites (exon skipping transcript preference for c.327-2G: 82.2 ± 9.7%, exon skipping transcript preference for c.538-1A: 67.9 ± 7.0%, Appendix A). While the skipping of exon 6 or exon 9 was predicted to result in an in-frame deletion, the alternative splicing events cause a frameshift and introduce a premature stop codon in the original open reading frame. All of these variants likely cause a loss of protein function.

### 2.3. Clinical Findings

Pregnancy and delivery were reported as uneventful in all of the affected individuals found to harbor novel disease-causing variants in *NDRG1*. The patients had normal motor milestones and started to walk between 1 year and 15 months. The mean age at onset was 8.0 ± 3.42 years, ranging between 7 and 12 years. The initial manifestations in all patients included distal muscle weakness in the lower limbs with impaired steppage gait and frequent falls. Weakness in the hands (in holding small objects) developed between the age of 10–17 years. The clinical findings are summarized in Table 2.

Neurological examination revealed severe distal muscle weakness in the lower limbs in all patients with muscle strength of toe flexors and extensors and plantar flexors and extensors of 0–1/5 in MRC score. In the upper limbs, the distal muscles were moderately to severely affected, with finger extensors being weaker than the flexor muscles (Table 2). The tendon reflexes were absent in the lower limbs in all of the patients and were preserved but attenuated in the upper limbs in 5/7. All patients had an unstable gait with marked steppage, except one case I-1, who had lost ambulation at the age of 23 years. The sensation in the distal parts of the four limbs produced by light touch, pin prick, and temperature sensations was diminished to absent in all of the affected, while vibration sense was impaired in the lower limbs in all patients but preserved in the upper limbs of 5/7. Muscle atrophy, involving abductor pollicis brevis, interosseous, gastrocnemius, and tibialis anterior muscles with formation of *pes cavus* and equinovarus and claw hands were present in all patients, with slight asymmetry between the left and right side in 2/5 (case II-1 and II-2). Scoliosis was observed only in one patient (case I-1).

Nerve conduction studies (NCS) of all of our patients were consistent with severe demyelination and secondary axonal degeneration. The results from NCS are presented in Table 3. SNAP from four limbs and CMAP from the lower ones could not be obtained from any of the affected individuals, except one (case IV-2), evaluated at the age of 2.5 years, when the CV of n. peroneus were found to be 12.8 m/s. In the upper limbs, CMAPs had severely reduced CV, prolonged distal latencies, and low amplitudes. In three of the patients (case I-1, case IV-1, and case IV-2), who were evaluated prospectively at two different ages, these parameters deteriorated throughout the disease evolution (Table 3). Needle EMG showed scattered fibrillation and neurogenic motor unit action potentials, which were more pronounced in the distal muscles. CMTNS ranged between 23 and 38 without a clear-cut correlation with age of examination.

In 4/7 patients affected, there were complaints of hearing loss. Interestingly, the two patients with no auditory complains (case II-1 and case II-2), aged 30 and 31 years, respectively, also had normal audiograms (Table 3). The audiograms of the rest were consistent with sensorineural hearing loss, especially for high frequencies. Brainstem auditory-evoked potentials showed increased latencies of waves I–III–V, as well as increased interpeak latencies in some patients (cases II-1, II-2, IV-1, IV-2, and IV-3). In the other 2/7 (cases I-1 and III-1), the impairment was so severe that it was not possible to identify the different waves on BAEP (Table 3).

## 3. Discussion

CMT4D was initially described in a Roma community located in Lom, a small town by the Danube River in northwest Bulgaria. Identification of additional families with the prevailing founder p.R148X mutation in the Roma community worldwide led to the impression that CMT4D is confined to this community. However, subsequent studies identified a handful of other mutations in singular families with no Roma ethnicity, thus demonstrating a wider geographical distribution [15,16,18,20,21,22,24]. Here, we report our findings of additional recurrent and novel founder variants in the *NDRG1* gene identified in Bulgarian CMT1 patients and provide functional evidence on the pathogenicity of novel splice-site variants in the gene in non-Roma patients.

We identified 11 families with likely disease-causing variants in the *NDRG1* gene in a cohort of 150 Bulgarian demyelinating CMT patients. This suggests a 7.3% frequency, which might be considered high for such a rare disease subtype. The patients we screened here were selected exclusively based on clinical criteria for CMT type 4 and also included families of Roma descent. As expected, the patients that carried the founder p.R148X mutation were of Roma descent, which is one reason for the overall high rate of *NDRG1* variants compared to previous studies. Six families carried this mutation in a homozygous state, while one family carried the recurrent p.R148X variant in compound heterozygosity with a novel frameshift variant (p.E170Gfs*35).

We report two splice-site variants in four families, one of which has previously been described in a Bulgarian non-Roma patient (c.538-1G>A). In contrast, the second variant is a novel finding. cDNA analysis from patient-derived biosamples revealed that the c.327-2G and c.538-1A alleles primarily result in exon skipping of exon 6 and exon 9, respectively, in the mature mRNA, causing an in-frame deletion in the putative protein. Amino acid residues in both exon 6 (ENSE00003463021) and exon 9 (ENSE00003493203) are relatively conserved in NDRG1, NDRG2b, NDRG3, and NDRG4 proteins, as well as in the *NDRG1* orthologues across different species [32]. Based to the crystal structure of the protein (PDB: 6ZMM), skipping exon 6 would abolish α3 and α4 helices, while skipping exon 9 abolishes the putative helix α6 and the α7 helix [32]. Both of these events likely cause a loss of protein function due to the disruption of the central α/β hydrolase domain. Interestingly, Pravinbabu et al. reported a splice-site variant (c.537+2_537+10del) in a 17-year-old Indian male. The authors show that the variant causes activation of a downstream cryptic splice site at intron 8, resulting in an insertion of 42 nucleotides into exon 8. This leads to an in-frame insertion of 14 amino acids between residues 179 and 180 that would alter the structure of the putative α6 helix [20]. The crystal structure of this helix is not resolved due to its high flexibility, which suggests that it allows for the binding of other proteins or substrates [32]; therefore, the authors argue that the alteration of the putative α6 helix may impact the interaction of other proteins with the NDRG1 protein [20].

Remarkably, the two splice-site variants (c.538-1G>A and c.327-2A>G) we report here are identified in four families that belong to an ethnic minority of Bulgarian Muslims. We showed a shared haplotype on the mutated allele between these families, suggesting an ancestral founder effect. The Bulgarian Muslim population is concentrated in the Rhodope Mountains in the southwest part of the country and is estimated to consist of about 160.000–240.000 individuals [33]. Our findings of new founder mutations in the *NDRG1* gene makes CMT4D the second most common autosomal recessive disorder described in this ethnic group after limb-girdle muscular dystrophy type 2G [34].

In the Bulgarian Muslim patients with the founder *NDRG1* variants, there was no delay in starting to walk, in contrast to some affected individuals previously described [2,15,16,27], but the initial complaints of distal weakness in the lower limbs and gait disturbances appeared in the first decade of life in six out of seven patients, which is in line with previous reports of patients carrying the Roma founder p.R148X variant [2], as well as other pathogenic variants [15,16,18,21]. In one patient (case IV-3), the initial symptoms occurred later at the age of 12 years, which was described in three other affected individuals [24], suggesting that sometimes the disease course can be milder. The patients reported herein showed other typical clinical features, including muscle weakness and wasting of the upper limbs, occurring in the second decade, tendon areflexia, skeletal and foot deformities. Although deafness is a common feature of CMT4D patients, observed in the second or third decade, not all CMT4D patients display hearing loss [12,24,35]. Strikingly, in two affected patients reported here, hearing complaints were not present and normal audiograms were found in the fourth decade. In the Roma group, Kalaydjieva et al. have described normal audiometric findings in only one patient out of nineteen in the fourth decade [2]. Their BAEP were consistent with increased latencies of waves I–III–V, as well as increased interpeak latencies, proving that auditory involvement can be subclinical, but still present. Nerve conduction studies of our patients showed a severe reduction in motor nerve conduction velocities in the upper limbs and unobtainable CMAP and sensory potentials, which is a usual finding in this disorder [2,9,11,12,13,21,25,26,27,28,29,35,36,37,38].

In conclusion, we conducted a large cohort screening for the *NDRG1* gene in patients with early onset, demyelinating CMT from Bulgaria and reported recurrent and novel likely disease-causing variants in this gene. We identified and functionally characterized two new founder mutations in an ethnic minority of Bulgarian Muslims causing the typical, albeit milder, clinical manifestation of CMT4D, where sensorineural hearing loss was absent. Together with the well-established p.R148X Roma founder mutation, the observation of two additional founder mutations in this gene likely contributes to the relatively high prevalence of this rare CMT subtype in the multiethnic Bulgarian population. Overall, our findings enhance the understanding of the diverse genetic and clinical characteristics present in CMT4D and provide insights into the diagnostic workup in genetically heterogeneous populations.

## 4. Materials and Methods

### 4.1. Patient Cohort

The study cohort consisted of 150 CMT type 1 index patients from distinct pedigrees living in Bulgaria. Written informed consent was obtained from all participants. The study complies with the ethical guidelines of the institutions involved.

The patients harboring novel likely disease-causing variants in *NDRG1* were interviewed to obtain information on ethnic origin, family history, age at onset, initial symptoms, distribution of muscle weakness and sensory disturbances, and disease progression, as well as current disability. The clinical examination performed for this report included standard neurological examination combined with detailed testing of muscle strength using the Medical Research Council (MRC) grading method [39] and evaluation of CMTNS-CMT neuropathy score (second version) [31].

Nerve conduction studies encompassed the evaluation of conduction velocities (CV) and distal latencies of median, ulnar, tibial, peroneal, and sural nerves and late responses (F and A waves). Sensory nerve action potentials were elicited antidromically. Needle electromyography of distal and proximal upper limb muscles (abductor digiti minimi, abductor pollicis brevis, biceps brachii, deltoid) and distal lower limb muscles (extensor digitorum brevis, abductor hallucis, tibialis anterior) was also performed.

All of the reported patients underwent standard audiometry and brainstem auditory-evoked potentials (BAEP). BAEP recordings were performed using click stimuli at 80 dB hearing pressure levels, with the amplifier band frequency set at 200 to 2000 Hz. The data were analyzed and compared to the normal values established in the respective electrophysiologic laboratories. The findings were interpreted as normal if they were within the range of the control value ±2 SD. In three patients (case I-1, case IV-1, and case IV-2), the auditory evaluations were performed twice during their routine follow-up period (ranging from 6 years to 34 years), providing additional information on the progression of the symptoms over time.

### 4.2. Genetic Studies

An amplicon target amplification assay (Agilent, Santa Clara, CA, USA) was designed for the multiplex amplification of all *NDRG1* targets (coding exons, intron-exon boundaries, and 5′UTR). Primers were designed using the mPCR software (Agilent). Specific target regions were amplified using multiplex PCR, followed by purification of the equimolar pooled amplicons using Agencourt AMPureXP beads (Beckman Coulter, Brea, CA, USA). Individual barcodes (Illumina Nextera XT) were incorporated in a universal PCR step prior to sample pooling. Libraries were sequenced on a MiSeq platform using a v2 reagent kit with a paired end read length of 250 bp (Illumina, San Diego, CA, USA). Co-segregation analysis of the variants with the disease was performed for all of the available family members using Sanger sequencing. Variants were classified for pathogenicity according to the guidelines recommended by the American College of Medical Genetics and Genomics [30].

Haplotyping was performed with publicly available STR markers flanking the *NDRG1* locus. An M13 tag (5′-AGCGGATAACAATTTCACACAGG-3′) was added to the 5′ end of the forward primers amplifying each marker. During the PCR reaction, a fluorescent M13-tagged FAM label was added, allowing fluorescent labeling of the amplification products. Fragments were size-separated on an ABI3730xl DNA Analyzer and further analyzed using the in-house TCA software package v.1 (Antwerp, Belgium). STR markers and the corresponding primer sequences are listed in Appendix A.

### 4.3. Functional Studies

Peripheral blood mononuclear cells were isolated on a Ficoll-Paque gradient, transformed with Epstein–Barr virus, and maintained as previously described [40]. Total RNA was purified from lymphoblast cells using the RNeasy Mini Kit (Qiagen, Hilden, Germany) and treated with DNase (TURBO DNA-free kit, Applied Biosystems, Waltham, MA, USA). Total RNA from blood samples (PAXgene Blood RNA Kit, Qiagen, Hilden, Germany) was extracted according to the manufacturer’s protocol. cDNA was synthesized using the SuperScriptTM III First-Strand Synthesis System (Invitrogen, Waltham, MA, USA) with random hexamers. The primers used for sequencing of the *NDRG1* transcripts are listed in Appendix A. The forward primer used for PCR was labeled with a FAM dye for allele quantification. A size-standard marker was added to the PCR products and run on ABI3730xl DNA Analyzer (Applied Biosystems) for size separation. The area under the peak of the electropherograms was calculated using the MAQ-S Software v.1.5.0 (Agilent). Additionally, the PCR products generated from the cDNA of the individuals were denatured and allowed to re-anneal to form heteroduplexes with the alternatively spliced gene products. T7-endonuclease treatment was performed on these heteroduplexes to cleave non-perfectly matched DNA fragments with 1U of the enzyme for 20 min at 37 °C and the DNA fragments were run on an agarose gel. In silico prediction for splicing impact was performed using the Alamut Visual Plus (v1.5.1) and SpliceAI (Broad Institute, Cambridge, MA, USA) tools.

## Figures and Tables

**Figure 1 ijms-25-09047-f001:**
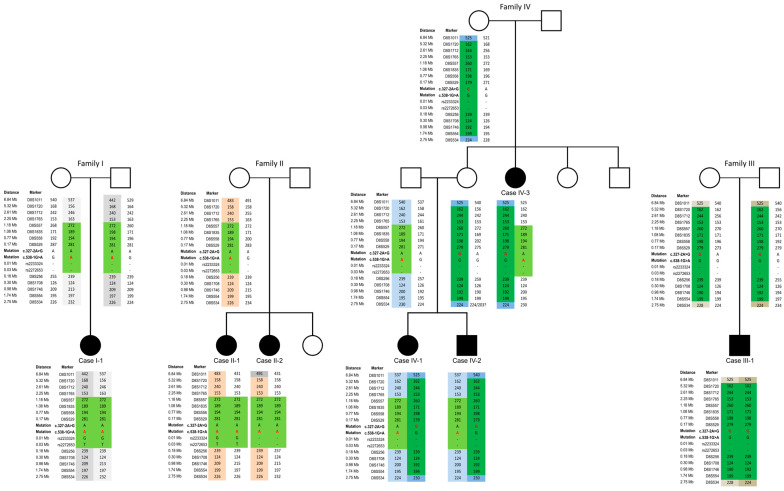
Haplotype schematics in families harboring the splice-site variants. The c.538-1G>A variant resides on a shared haplotype block of 2.4 Mb between the three families (I, II, and IV), shown in light green, and the c.327-2G variant resides on a 9.6 Mb haplotype block shared between families III and IV, shown in dark green. The individual alleles can be followed by additional colors in each pedigree. The splice site variants are denoted in red letters. Circles represent females, squares represent males, filled in shapes represent affected individuals.

**Figure 2 ijms-25-09047-f002:**
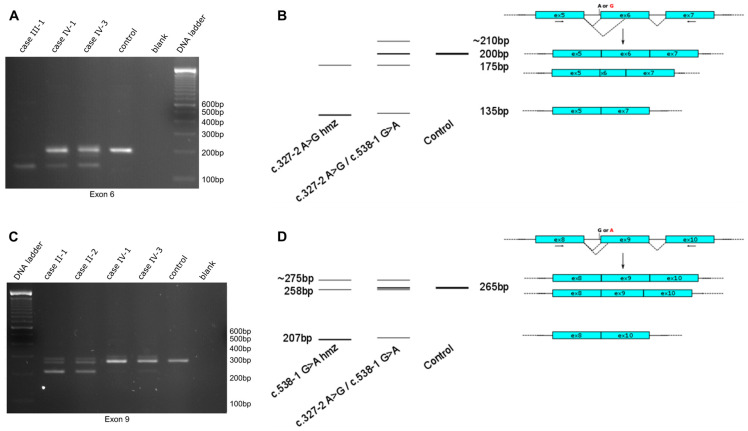
Splicing analysis of the c.327-2A>G and c.538-1G>A alleles. Pathogenic variants are denoted with red letters. (**A**) An amount of 2.5% agarose gel ran for 3 h at 100V, showing amplification of cDNA with primers flanking exon 6 of the *NDRG1* gene; (**B**) graphical overview and interpretation of (**A**); (**C**) 2.5% agarose gel ran for 3 h at 100V, showing amplification of cDNA with primers flanking exon 9 of the *NDRG1* gene; (**D**) graphical overview and interpretation of (**C**). Cyan boxes represent exons, arrows represent the primer positions used for the analyses.

**Table 1 ijms-25-09047-t001:** Genetic findings of patients with CMT4D/HMSN-Lom reported in this study.

Family	Individual	Ethnicity	Variant in *NDRG1*	Zygosity	ACMG Criteria	ACMG Classification	Interpretation
I *	I-1	Bulgarian Muslims	c.538-1G>A	Homozygous	PVS1, PS3, PM2	Pathogenic	Novel splice-site variant
II	II-1	Bulgarian Muslims	c.538-1G>A	Homozygous	PVS1, PS3, PM2	Pathogenic	Novel splice-site variant
	II-2		c.538-1G>A	Homozygous	PVS1, PS3, PM2	Pathogenic	Novel splice-site variant
III	III-1	Bulgarian Muslims	c.327-2A>G	Homozygous	PVS1, PS3, PM2	Pathogenic	Novel splice-site variant
IV	IV-1	Bulgarian Muslims	c.538-1G>A	Compound heterozygous	PVS1, PS3, PM2	Pathogenic	Novel splice-site variant
			c.327-2A>G		PVS1, PS3, PM2	Pathogenic	Novel splice-site variant
	IV-2		c.538-1G>A	Compound heterozygous	PVS1, PS3, PM2	Pathogenic	Novel splice-site variant
			c.327-2A>G		PVS1, PS3, PM2	Pathogenic	Novel splice-site variant
	IV-3		c.538-1G>A	Compound heterozygous	PVS1, PS3, PM2	Pathogenic	Novel splice-site variant
			c.327-2A>G		PVS1, PS3, PM2	Pathogenic	Novel splice-site variant
V	V-1	Roma	c.508dup, p.E170Gfs*35	Compound heterozygous	PVS1, PM2, PM3	Pathogenic	Novel frameshift variant
			c.442C>T, p.R148X		PVS1, PS3, PM2	Pathogenic	Recurrent disease-causing variant
VI	VI-1	Roma	c.442C>T, p.R148X	Homozygous	PVS1, PS3, PM2	Pathogenic	Recurrent disease-causing variant
VII	VII-1	Roma	c.442C>T, p.R148X	Homozygous	PVS1, PS3, PM2	Pathogenic	Recurrent disease-causing variant
VIII	VIII-1	Roma	c.442C>T, p.R148X	Homozygous	PVS1, PS3, PM2	Pathogenic	Recurrent disease-causing variant
IX	IX-1	Roma	c.442C>T, p.R148X	Homozygous	PVS1, PS3, PM2	Pathogenic	Recurrent disease-causing variant
	IX-2		c.442C>T, p.R148X	Homozygous	PVS1, PS3, PM2	Pathogenic	Recurrent disease-causing variant
X	X-1	Roma	c.442C>T, p.R148X	Homozygous	PVS1, PS3, PM2	Pathogenic	Recurrent disease-causing variant
XI	XI-1	Roma	c.442C>T, p.R148X	Homozygous	PVS1, PS3, PM2	Pathogenic	Recurrent disease-causing variant

* Family previously reported in Hunter et al. 2003 [21], *Hum Mutat*.

**Table 2 ijms-25-09047-t002:** Clinical features of Bulgarian Muslim patients with CMT4D/HMSN-Lom reported in this study.

Case ID	Age at Starting to Walk	Age at Onset	Initial Symptoms	Age of Involvement of Upper Limbs	Age at Assessment	Tendon Reflexes					Superficial Sensation		Deep Sensation		Limb Deformities		Hearing Loss	CMTNS
						Biceps	Triceps	Brachioradial	Knee	Achille	Upper Limbs	Lower Limbs	Upper Limbs	Lower Limbs	Upper Limbs *	Lower Limbs **		
I-1	NA	7 y	Impaired gait, distal weakness and impaired sensation in LL	10 y	43 y	-	-	-	-	-	D	D	D	D	+	+ Pes equinovarus, scoliosis	+	38
II-1	1 y 3 mo	7 y	Weakness in distal parts of lower limbs	12 y	30 y	+	+	-	-	-	D	D	N	D	+ R>L	+	-	23
II-2	1 y 4 mo	8 y	Weakness in distal parts of lower limbs	13 y	31 y	+	+	-	-	-	D	D	N	D	+ R>L	+ The right limb is 7 cm shorter due to congenital luxation of right art. coxae	-	28
III-1	1 y	9 y	Weakness in the lower limbs, impaired gait	16 y	20 y	+	+	-	-	-	D	D	N	D	+	+ Pes equinovarus	+	24
IV-1	1 y	7 y	Frequent falls, impaired gait	11 y	15 y	+	+	-	-	-	D	D	N	D	+	+	+	34
IV-2	1 y 2 mo	6 y	Frequent falls, impaired gait	10 y	13 y	+	+	-	-	-	D	D	N	D	+	+	+	29
IV-3	1 y 2 mo	12 y	Weakness in distal parts of lower limbs	16 y	36 y	-	-	-	-	-	D	D	D	D	+	+ Pes equinovarus, more severe on the right	+	34

Abbreviations: N: normal; D: decreased; R: right; L: left; ND: not done; NA: not available; CMTNS: CMT neuropathy score second version [31]. * Upper limb deformities encompass the following: hypotrophies of distal forearm, m. thenar, hypothenar, mm. interossei, flexion contractures of the fingers. ** Lower limb deformities encompass the following: Achille tendon contractures, pes cavus.

**Table 3 ijms-25-09047-t003:** Nerve conduction studies and auditory evoked potentials of Bulgarian Muslim patients with CMT4D/HMSN-Lom reported in this study.

Case	N. medianus CV/DL/A	N. ulnaris CV/DL/A	N. peroneus CV/DL/A	N. tibialis CV/DL/A	Audiometry	Auditory Evoked Potentials					
						Latencies ms			Interpeak Latencies ms		
						Wave I R/L	Wave III R/L	Wave V R/L	I-III R/L	III-V R/L	I-V R/L
Case I-1 at 9 years	NA	10.3/NA/NA	NM	NA	Severe sensorineural hearing loss	NA					
Case I-1 at 43 years	NM	NM	NM	NM	Severe sensorineural hearing loss	Severely affected. Not possible to identify the different waves.					
Case II-1 at 30 years	9.7/8.6/1.2	11.5/9.2/0.3	NM	NM	Normal	1.9/2.0	4.8/5.0	7.5/7.8	2.9/3.0	2.7/2.8	5.6/5.8
Case II-2 at 31 years	16.7/11.9/0.27	12.4/9.51/0.34	NM	NM	Normal	1.9/1.6	4.4/4.4	7.2/7.1	2.5/2.8	2.8/2.7	5.3/5.5
Case III-1 at 20 years	10.8/23.8/1.8	12.5/21.3/1.9	NM	NM	Severe sensorineural hearing loss	Severely affected. Not possible to identify the different waves.					
Case IV-1 at 9 years	13.7/11.7/2.8	NA	NM	NM	ND	ND	ND	ND	ND	ND	ND
Case IV-1 at 15 years	NM	NM	NM	NM	Sensorineural hearing loss, especially for high frequencies	2.96/2.87	6.04/6.00	8.84/8.46	3.08/3.13	2.8/2.46	5.88/5.59
Case IV-2 at 2.5 years	15.6/10.4/2.2	NA	12.8/12.5/0.2	NA	ND	ND	ND	ND	ND	ND	ND
Case IV-2 at 12 years	5.8/13.8/0.1	NM	NM	NM	Sensorineural hearing loss, especially for high frequencies	3.05/3.07	5.87/5.98	7.95/7.99	2.82/3.91	2.08/2.01	4.9/4.92
Case IV-3 at 36 years	NM	NM	NM	NM	Sensorineural hearing loss, especially for high frequencies	2.1/2.2	4.8/5.0	7.7/8.0	2.7/2.8	2.9/3.0	5.6/5.8

CV: conduction velocity measured in m/s; DL: distal latency measured in ms; A: amplitude measured in mV; NA: not available; ND: not done; NM: not measurable, SNAP were not measurable, the data are on CMAP.

## Data Availability

The original contributions presented in the study are included in the article/Appendix A, further inquiries can be directed to the corresponding authors.

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
