# Peer review of "Identification and Characterization of Novel Founder Mutations in NDRG1: Refining the Genetic Landscape of Charcot–Marie–Tooth Disease Type 4D in Bulgaria"

_ijms, 2024, doi:10.3390/ijms25169047_

Round 1

Reviewer 1 Report

Comments and Suggestions for Authors

The authors present genetic and phenotypic data from a large Bulgarian cohort of patients with a recessive form of demyelinating Charcot-Marie-Tooth disease (CMT) caused by loss-of-function mutations in the N-Myc Downstream Regulated 1 (NDRG1) gene. This study revealed 2 novel splice-site variants exclusively found in Bulgarian Muslims, which abolish the natural splice acceptor sites for intron 5 and intron 8. These variants reside in shared haplotypes of 2.4Mb and 9.6Mb between multiple families indicating an ancestral founder. Clinical assessment revealed absent lower limb tendon reflexes in all patients and absent upper limb tendon reflexes in 5/7 patients, hearing loss in 5/7 patients, and upper and lower limb deformities in all patients. Overall the data is clearly presented and well described. Some minor suggestions for the authors are below.

Figure 1 was a tiny bit confusing. It would be helpful (to me) if the disease-associated haplotype blocks that are highlighted in green for affected individuals were also shown in the carriers (either with a similar color or just an outline). Adding an explanation of the colors used in the Figure caption would also be useful (i.e. stating that the green areas indicate disease-associated haplotype blocks).

In Figure 2, could the authors graph their quantification of splicing preference for splice-site variants?

Could the authors add a little more detail to the gene diagrams in Figure 2B and 2D showing which nucleotides are excluded in the splice variants? Or provide this information in the supplemental material?

Author Response

The authors present genetic and phenotypic data from a large Bulgarian cohort of patients with a recessive form of demyelinating Charcot-Marie-Tooth disease (CMT) caused by loss-of-function mutations in the N-Myc Downstream Regulated 1 (NDRG1) gene. This study revealed 2 novel splice-site variants exclusively found in Bulgarian Muslims, which abolish the natural splice acceptor sites for intron 5 and intron 8. These variants reside in shared haplotypes of 2.4Mb and 9.6Mb between multiple families indicating an ancestral founder. Clinical assessment revealed absent lower limb tendon reflexes in all patients and absent upper limb tendon reflexes in 5/7 patients, hearing loss in 5/7 patients, and upper and lower limb deformities in all patients. Overall the data is clearly presented and well described. Some minor suggestions for the authors are below.

Figure 1 was a tiny bit confusing. It would be helpful (to me) if the disease-associated haplotype blocks that are highlighted in green for affected individuals were also shown in the carriers (either with a similar color or just an outline). Adding an explanation of the colors used in the Figure caption would also be useful (i.e. stating that the green areas indicate disease-associated haplotype blocks).

We would like to thank the Reviewer for this helpful suggestion. We updated the figure to show the haplotype blocks in carriers/ In the figure legend we explained what each color represents, please see line 115.

In Figure 2, could the authors graph their quantification of splicing preference for splice-site variants?

Could the authors add a little more detail to the gene diagrams in Figure 2B and 2D showing which nucleotides are excluded in the splice variants? Or provide this information in the supplemental material?

We now added a new supplementary figure (Figure S4) to show the nucleotide sequence and the splicing events of the amplicons analyzed in Figure 2. This figure also includes a quantification graph to show percent allele preference due to the two splice site variants.

Reviewer 2 Report

Comments and Suggestions for Authors

In this manuscript, the authors characterize a cohort of patients harboring novel NDRG1 mutations among a population of Bulgarian patients with demyelinating Charcot-Marie-Tooth 4D disease.  Analysis of the transcripts resulting from the mutant genes suggest that the mutations are loss of function alleles.  This manuscript is well written and contributes to our understanding of the mutational landscape of CMT. One issue described below should be addressed, after which I think the manuscript should be suitable for publication:

Line 153 "variants likely cause a possible protein loss of function." No need for the word possible if you already have the qualifier "likely," but could this statement be made a bit more confidently based on what is already know about the gene?  What is known about the function of the protein to make the case more strongly? Are functional domains absent?  How does the theoretical protein compare to those produced by previously identified or characterized mutant alleles? Is the c.538 mutation functionally equivalent to the already characterized c.537 mutation described in the introduction?

Author Response

In this manuscript, the authors characterize a cohort of patients harboring novel NDRG1 mutations among a population of Bulgarian patients with demyelinating Charcot-Marie-Tooth 4D disease.  Analysis of the transcripts resulting from the mutant genes suggest that the mutations are loss of function alleles.  This manuscript is well written and contributes to our understanding of the mutational landscape of CMT. One issue described below should be addressed, after which I think the manuscript should be suitable for publication:

Line 153 "variants likely cause a possible protein loss of function." No need for the word possible if you already have the qualifier "likely," but could this statement be made a bit more confidently based on what is already know about the gene?  What is known about the function of the protein to make the case more strongly? Are functional domains absent?  How does the theoretical protein compare to those produced by previously identified or characterized mutant alleles? Is the c.538 mutation functionally equivalent to the already characterized c.537 mutation described in the introduction?

We would like to thank the Reviewer for this suggestion. We corrected the statement in line 153 (currently in line 154). We also extended the third paragraph of the discussion and included information regarding the crystal structure of NDRG1 and what kind of alterations the splice site variants might be causing in the protein structure. As kindly suggested by the Reviewer, we included a comparison of our findings to the findings and discussion of the paper by Pravinbabu et al. (2024) which reports another splice site variant (c.537+2_537+10del).